# Possible Correlation between Urocortin 1 (Ucn1) and Immune Parameters in Patients with Endometriosis

**DOI:** 10.3390/ijms24097787

**Published:** 2023-04-24

**Authors:** Monika Abramiuk, Karolina Frankowska, Krzysztof Kułak, Rafał Tarkowski, Paulina Mertowska, Sebastian Mertowski, Ewelina Grywalska

**Affiliations:** 1Independent Laboratory of Minimally Invasive Gynecology and Gynecological Endocrinology, Department of Oncological Gynaecology and Gynaecology, Medical University of Lublin, Staszica 16 St., 20-081 Lublin, Poland; 21st Chair and Department of Oncological Gynaecology and Gynaecology, Student Scientific Association, Medical University of Lublin, Staszica 16 St., 20-081 Lublin, Poland; k.frankowska10@gmail.com; 31st Chair and Department of Oncological Gynaecology and Gynaecology, Medical University of Lublin, Staszica 16 St., 20-081 Lublin, Poland; krzysztof.kulak@gmail.com (K.K.); rafaltar@yahoo.com (R.T.); 4Department of Experimental Immunology, Medical University of Lublin, Chodźki 4a St., 20-093 Lublin, Poland; paulina.mertowska@umlub.pl (P.M.);

**Keywords:** endometriosis, urocortin 1, deep infiltrating endometriosis, immune system, interleukins, ovarian endometrioma, peritoneal endometriosis

## Abstract

The etiology of endometriosis (EMS) has not been clearly elucidated yet, and that is probably the reason why its diagnostic process is frequently long-lasting and inefficient. Nowadays, the non-invasive diagnostic methods of EMS are still being sought. Our study aimed to assess the serum and peritoneal fluid levels of urocortin 1 (Ucn1) in patients with EMS and healthy women. Moreover, considering the immune background of the disease, the association between Ucn1 and several immune parameters was studied in both groups. We found that the serum Ucn1 level was significantly upregulated in women with EMS compared to healthy patients. Moreover, higher serum Ucn1 levels tended to correspond with more advanced stages of the disease (*p* = 0.031). Receiver operating characteristic (ROC) analysis revealed that based on serum Ucn1 levels, it is possible to distinguish deep infiltrating endometriosis (DIE) from among other EMS types. Together, these results indicate Ucn1 as a possible promising biomarker of EMS: however, not in isolation, but rather to enhance the effectiveness of other diagnostic methods.

## 1. Introduction

Endometriosis (EMS) is a chronic estrogen-dependent disease characterized by pathological implantation and the growth of endometrial tissue outside the uterine cavity. Endometrial changes are most often located on the surface of the ovaries in the form of ovarian cysts and on the surface of the peritoneum of the pelvis as well as within the deep structures of the pelvis. Literature data indicate three main types of EMS, which include: ovarian endometriosis (OMA), peritoneal endometriosis (PE) and deep infiltrating endometriosis (DIE) [1,2,3,4]. The most characteristic symptoms of EMS, such as chronic pelvic pain, dysmenorrhea, and dyspareunia, cause a significant reduction in the quality of life of patients [5,6]. Epidemiological data estimate that EMS affects approximately 10% of women of childbearing age [7,8]; however, its prevalence among women struggling with infertility reaches up to 50%, which means that these two conditions are closely related [8,9]. Currently, there are several theories trying to explain the mechanisms of EMS formation. Scientists suggest the participation of many genetic, environmental and immunological factors involved in the pathogenesis of this disease [10,11,12]. Currently, the gold standard for EMS diagnosis is vaginal examination combined with imaging tests (ultrasound or magnetic resonance imaging) without mandatory diagnostic laparoscopy (LPS), which is another diagnostic option [13]. However, the used diagnostic methods are quite invasive, and scientists and clinicians are increasingly looking for new non-invasive biomarker molecules that may be included in the diagnostic path in the future [14]. One such molecule may be urocortin 1 (Ucn1). It is a 40-amino acid neuropeptide that belongs to the CRF family and acts through the corticotropin-releasing factor-1 (CRFR-1) and corticotropin-releasing factor receptor-2 (CRF-2) receptors, to which it has similar affinity [15,16,17]. Although Ucn1 was originally described as a peptide associated with brain tissue, intensive research has also shown its presence in many cells, tissues and organs, both in animal and human samples [18]. As indicated in the literature, Ucn1 as a neuropeptide is mainly involved in the regulation of the nervous system on many levels as well as playing a significant role in the control of the cardiovascular system [19,20]. In addition, scientists have shown that Ucn1 is also associated with the immune system, where it plays a key role in the immunomodulation of T lymphocytes, B lymphocytes, macrophages, and monocytes. According to the current state of knowledge, Ucn1 can be considered both as a peptide with anti-inflammatory properties or as an inhibitor of autoimmune processes [21,22]. The wide distribution and multiple mechanisms of action of Ucn1 are also specific to the female reproductive system [22,23]. Through its autocrine and paracrine actions, Ucn1 is involved in the entire spectrum of obstetric conditions, including embryo implantation, the pathogenesis of miscarriage, the onset of labor (including premature birth) and the pathogenesis of pre-eclampsia. What is more, Ucn1 has been shown to be also produced by ovarian follicles; hence, researchers attribute to it participation in steroidogenesis [22,23]. There is also increasing scientific evidence that Ucn1 may play an active role in endometrial function as endometrial cells are one of the sources of Ucn1 [24,25]. Studies have also shown that Ucn1 secretion varies depending on the phase of the menstrual cycle, as conditions of high progesterone result in an increased production of peptides [26]. Moreover, Ucn1 mediates the growth of endometrial cells and plays a major role in the functional and morphological changes that occur in the endometrium during the decidualization process [27]. 

Thus, the question arises as to whether the action of Ucn1 coincides with the etiology of pathophysiological conditions affecting the endometrium, including the development of endometriosis.

Our study aims to determine whether there is a relationship between the concentration of Ucn1 in serum and peritoneal fluid and various types and stages of endometriosis as well as its development. In addition, bearing in mind the immunological background of endometriosis, as well as the immunomodulatory effect of Ucn1 on cells of the immune system, we decided to check the existence of a relationship between Ucn1 and various subpopulations of T and B lymphocytes and the cytokines they produce.

## 2. Results

### 2.1. Basic Characteristics of Patients Diagnosed with Endometriosis and Healthy Volunteers Included in the Study

The study group consisted of 76 patients with histopathologically confirmed EMS. The degree of differentiation of patients in particular stages of EMS is presented in Figure 1A. Among all enrolled patients, 30 of them had OMA (39.48%), 11 DIE (14.47%) and 35 PE (46.05%) (Figure 1B). In addition, 57.90% of the patients had adhesions, 50% were infertile, and 80.26% of the patients reported pelvic pain (Figure 1C).

The mean age of patients in this group was 35 years ± 6.14. The control group consisted of 20 healthy women who were age-matched to the patients in the study group (36.65 ± 6.69). Table 1 presents the results of basic peripheral blood count parameters and biochemical tests performed for both groups of patients before the procedure. There were no statistically significant differences in most of the analyzed parameters, with the exception of the white blood cells and neutrophils levels, which were higher in patients from the study group compared to control patients (Table 1).

### 2.2. Evaluation of Ucn1 Concentration in the Serum and Peritoneal Fluid of Patients with EMS and the Control Group and Its Possible Use as a Biomarker Molecule in the Diagnosis of EMS

In the next stage of the study, the level of Ucn1 in serum and peritoneal fluid in EMS patients was evaluated. EMS patients showed a more than nine times higher level of Ucn1 in the serum than in the peritoneal fluid, and the achieved values were almost three times higher than in healthy patients, which may indicate the involvement of Ucn1 in the pathogenesis of EMS (Table 2). The obtained test results were compared with the levels of two other tumor markers: CA 125 (carcinoma antigen 125), which is a marker of ovarian cancer that is useful in assessing the effectiveness of treatment, in detecting relapses and predicting the survival time of patients, and HE4 (human epididymis protein 4), which in laboratory practice is used in the diagnosis and differentiation of epithelial ovarian cancer. Patients diagnosed with EMS had a 4-fold higher level of CA 125 than patients from the control group, while in the case of HE4 determinations, no statistically significant differences were noted between the two study groups (Table 2).

The obtained results suggest that CA 125 and Ucn1 can be considered as potential biomarker molecules for the development of endometriosis. Therefore, we conducted an analysis to assess the effectiveness of the selected molecules as potential biomarker molecules. From the data in Table 3, CA 125 was the best marker for EMS differentiation, and serum Ucn1 was slightly less effective in Table 3.

### 2.3. Evaluation of the Effectiveness of Serum Ucn1, CA 125, HE4 and Peritoneal Ucn1 as a Potential Biomarker Molecules of Particular EMS Types

In the next stage of our analysis, we decided to check whether there are differences between the serum levels of Ucn1, CA 125 and HE4 in individual subgroups of patients with EMS. For this purpose, we conducted a detailed assessment of the concentration of selected marker molecules among patients with different EMS types. The detailed results are presented in Table 4.

Figure 2 shows the ROC analysis for the serum Ucn1 concentration used as a discriminator of the presence of different EMS types (OMA, DIE and PE). The AUC values for OMA and PE do not provide discrimination, i.e., the ability to diagnose patients with or without these disease types based on the estimated cutpoints, AUC = 0.51 and AUC = 0.54, respectively. For DIE, the AUC ranging 0.71 can be considered as acceptable for distinguishing this EMS type from among others.

The AUC values for Ucn1 peritoneal fluid concentrations do not indicate discrimination for any of the OMA, DIE, PE, i.e., the ability to diagnose patients with or without any disease type based on the estimated cut point. The ROC curves for each EMS type can be found in Figure 3.

### 2.4. Assessment of the Relationship between Ucn1 Concentrations in a Group of Patients with EMS and the Stage of the Disease

Correlation analysis showed a significant relationship between serum Ucn1 concentration and the severity of EMS, τb = 0.22, z = 2.16, *p* = 0.031. A positive correlation coefficient with moderate effect size indicated that serum urocortin 1 moderately increased with increasing EMS stage. On the other hand, the conducted correlation analysis showed a non-significant negative relationship with a small effect between the concentration of urocortin 1 in the peritoneal fluid and the EMS stage, τb = −0.02, z = −0.23, *p* = 0.816.

### 2.5. Evaluation of Selected Parameters of the Functioning of the Immune System in the Course of EMS in Relation to Healthy Volunteers

The next stage of the research was to determine the functioning of the immune system of EMS patients in relation to the control group. For this purpose, we analyzed the percentage of T lymphocytes: helper (Th), cytotoxic (Tc) and regulatory (Treg), as well as B lymphocytes in the peripheral blood of recruited patients. In addition, we evaluated the expression of CD25 and CD69 activation markers. The CD25 marker is present on the surface of maturing T and B lymphocytes (not found on naive lymphocytes) and transiently expressed on activated T and B lymphocytes, while it is constitutively expressed on Treg lymphocytes. On the other hand, CD69 is considered a classical early marker of lymphocyte activation. The obtained test results are summarized in Table 5. As we can see, patients diagnosed with EMS are characterized by a higher percentage of CD3+ lymphocytes alone and activated T and B lymphocytes (expressing both CD25+ and CD69+) in all analyzed subpopulations compared to patients from the control group. This may indicate dysregulation of the functioning of the immune system in the course of EMS.

A detailed analysis of selected immunological parameters in individual EMS subtypes also provided valuable information, which is presented in Table 6. Additionally, an exemplary analysis of the percentage of lymphocytes performed by flow cytometry is shown in Figure 4 and Figure 5.

As the collected data indicate, there are a number of significant statistical differences between the individual EMS types and the control group. Furthermore, we also demonstrated several significant changes in immune function between the EMS subtypes themselves. In the case of OMA and DIE, we can see an increase in the percentage of CD4+/CD25/Foxp3+ T lymphocytes, CD3+/CD25+ as well as CD4+/CD25+ T lymphocytes and CD19+/CD25+ B lymphocytes for patients with DIE. A higher percentage for patients with OMA in relation to DIE concerned CD3+/CD69+, CD4+/CD69+ and CD8+/CD69+ T lymphocytes. Statistically significant differences between PE and DIE concerned the increased percentage of CD4+/CD25/Foxp3+ T lymphocytes, CD4+/CD25+ T lymphocytes, as well as CD19+/CD25+ B lymphocytes, in patients diagnosed with DIE compared to PE. PE patients had an increase in the incidence of CD3+/CD8+ and CD3+/CD69+ as well as CD4+/CD69 and CD8+/CD69+ T lymphocytes.

The next step of our research was the quantitative determination of the concentration of selected cytokines IL-2, IL-4, IL-6, IL-10, and IFN-γ in the serum of all patients included in the study and in the case of EMS patients, also in the peritoneal fluid. Serum levels of all analyzed cytokines, except for IL-4, were higher in EMS patients than in controls (Table 7). In the case of IL-6, this increase was over 7-fold, while for IL-10, it was over 2.5-fold, for IL-2, it was over 6-fold, and for IFN-γ, it was almost 3-fold compared to the values recorded among healthy patients (Table 7). Particularly noteworthy are the values of cytokines recorded in the peritoneal fluid of patients with EMS, where for IL-6, the increase in mean values observed was 6.41 times higher than in serum. Similar relationships were also noted for IL-10, an increase of 4.05 times, while for IFN-γ, it was as much as 17.42 times higher than in serum (Table 7).

We also looked in detail at the concentration of cytokines in individual EMS subtypes. According to the collected data, the level of the analyzed cytokines differs between the studied EMS subtypes (especially for the concentration of IL-2 within all subtypes) (Table 8). Moreover, statistically significant differences were also noted for the concentration of IL-6 and IL-10 between PE and DIE and between OMA and PE (Table 8).

### 2.6. Assessment of the Relationship between Urocortin Concentration and Selected Parameters of the Functioning of the Immune System of Patients with Endometriosis

This section discusses the correlation coefficients and results regarding the significance of the relationship between serum Ucn1 concentration and selected parameters of the functioning of the immune system for a group of patients with endometriosis and female patients. In the EMS group, an increase in the concentration of urocortin in the serum was associated only with a statistically significant increase in the percentage of CD19+ B lymphocytes and the concentration of Ucn1 in the peritoneal fluid (0.421) and the level of CA 125 (0.311). Moreover, a statistically significant negative correlation was found between the concentration of Ucn1 in the serum and the concentration of IFN-γ in the peritoneal fluid (−0.301). Correlation analysis of individual subpopulations of lymphocytes showed positive correlations between the level of CD3+ lymphocytes and IL-2 in the peritoneal fluid (0.336) and the percentage of regulatory T lymphocytes (0.303). In the case of B lymphocytes, their percentage correlated positively with the percentage of CD19+/CD25+ lymphocytes (0.414) and negatively with the level of IL-6 in the peritoneal fluid (−0.349). The level of Ucn1 in the peritoneal fluid significantly correlated with the percentage of B lymphocytes in the peripheral blood of patients with EMS (0.365). We observed additional relationships for the percentage of TCD4+ lymphocytes with the percentage of CD8+ T cells (−0.568) and the level of IL-4 in the peritoneal fluid (0.282). The most numerous correlations concerned the percentage of CD8+ T lymphocytes in peripheral blood with the percentage of CD8+/CD25+ lymphocytes (−0.350); CD3+/CD69+ (0.425); CD4+/CD69+ (0.425); and CD8+/CD25+ (0.425) as well as serum IL-6 (−0.323) and peritoneal fluid IL-4 (−0.405).

## 3. Discussion

### 3.1. Ucn1 as a Potential EMS Biomarker

Currently, according to the ESHRE guidelines, no biomarkers, measured in serum, urine or peritoneal fluid, are recommended for EMS diagnosis [13]. Nevertheless, the utility of various non-invasive diagnostic methods is still discussed in the literature, with the hope of developing non-invasive biomarker-based assays helping to detect EMS [28]. Thus, we assume that investigation of the Ucn1 role in EMS should aim at the same—not establishing it as a single independent diagnostic factor for usage in isolation but considering it as a one component of the complex diagnostic process.

The important finding of our study is that both CA 125 and Ucn1 achieved higher serum concentrations in women with EMS compared to controls, meanwhile pointing to the greater CA 125 role. Observations made by Maia et al., regarding the Ucn1 diagnostic value, were to a large extent consistent with our results. They measured serum Ucn1 levels in patients with EMS and in healthy controls and noted that in the first group, the obtained values were substantially higher. In addition, as in our study, the authors also indicated a Ucn1 serum concentration value that is able to distinguish the presence and absence of EMS. However, they proposed a cutoff point which was significantly lower than ours: 0.046 ng/mL vs. 5.94 ng/mL, respectively. This difference is difficult to explain when considering similar AUC values in both cases (AUC = 0.827; AUC = 0.850) [29]. Nevertheless, it is worth noting that contrastingly to us, the authors did not present the data regarding the stage of the disease, and hence, differences on this issue between our studies may be a source of above-mentioned divergences [29].

On the other hand, Kempuraj et al. decided to evaluate the Ucn1 levels in the tissue from endometriotic lesions and normal endometrium of women with EMS as well as in the endometrium of healthy controls. They found that while the ectopic endometrium was characterized by high Ucn1 expression, the eutopic endometrium of women with EMS showed such expression at a much lower level. Such results shed light on the importance of the material in which the peptide concentration is determined and show that the value of Ucn1 concentration patterns will not be the same for different tissues [30].

### 3.2. Evaluation of Ucn1 as a Beneficial Tool Able to Distinguishing Different EMS Types

Moreover, our study demonstrated that it is not possible to confirm or exclude the presence of ovarian endometriotic cysts (OMA), based on the levels of Ucn1, either measured in plasma or peritoneal fluid. On the other hand, currently available studies have attempted to answer the question, whether there is a possibility to distinguish between OMA and other benign non-endometriotic ovarian lesions. However, the reports of this issue are largely inconsistent. Florio et al. conducted a study on the research group including women with OMA, which was compared to the control group consisting of women with other benign ovarian cysts. Their observations gave us encouraging conclusions about the use of serum Ucn1 measurements in differentiating various types of ovarian lesions [31]. Nevertheless, Chmaj-Wierzchowska et al. [32] and Tokmak et al. [33], who also sought to compare the Ucn1 levels between women with OMA and women with ovarian lesions characterized with other etiology, did not support aforementioned results. In both studies, no differences in Ucn1 serum concentration between these two groups were noticed. In addition, the study investigating the association between Ucn1 expression with recurrence of OMA showed that this peptide did not reflect such risk. Thus, Yalcin et al. agreed that in general, Ucn1 should not be implemented in broadly understood OMA evaluation [34].

When in the course of our study, we analyzed individual types of EMS, we observed that for OMA, as well as for PE, serum Ucn1 concentrations were not specific, and its values should not be considered as the markers of the occurrence of these disease types. However, things are different when it comes to the DIE cases—we noted that higher serum Ucn1 levels have predisposed for receiving such a diagnosis. In this regard, the observations obtained by Carrarelli et al. stand in line with our assumption. The authors investigated how Ucn1 mRNA expression behaves in lesions classified as OMA and DIE. Their main highlight was that Ucn1 mRNA expression was much elevated in DIE lesions compared to OMA [35]. Our observations concerning DIE distinguishing seem to have essential clinical relevance, especially considering the fact that in imaging examinations, this EMS type is sometimes erroneously diagnosed as other non-gynecological conditions [36]. Thus, the combination of such a serum biomarker with the usage of novel standardized criteria of imaging studies [37] or newly modified imaging techniques [38] may result in positive diagnostic outcomes of patients affected by DIE.

Further, Florio et al. have compared a group of women who had only OMA with a group who had both OMA and PE, and consistent with our findings, they described that there were no differences in serum Ucn1 levels between these disease types. Thus, they confirmed the meaninglessness of the use of Ucn1 in the differentiation of ovarian and peritoneal EMS manifestations [31].

Continuing to focus on distinct types of EMS, we observed that for OMA, DIE and PE peritoneal fluid, Ucn1 concentrations were not effective in predicting their occurrence. This again proved that conclusions drawn based on the blood Ucn1 measurements should not be considered in advance to be identical to Ucn1 measurements made in other samples.

### 3.3. The Role of Ucn1 in Distinguishing of Different Stages of EMS

However, we believe that the most meaningful point coming from our study concerns the relationship between serum Ucn1 concentrations and stages of EMS. We found that higher plasma Ucn1 levels were characteristic for more advanced stages of EMS. Although previously conducted studies did not investigate the link between Ucn1 and EMS staging, there are several possible explanations for this relationship. Firstly, Ucn1 promotes the enhanced expression of matrix metalloproteinase-9 (MMP-9) [39]—the molecule responsible for endometrial cells spreading [40], which amount also increases with the EMS stage [41]. Such greater spreading of endometrial cells guarantees enhanced invasion of the disease and as a consequence results in its more advanced stages. Another possible reason for such dependence may be related to the increased interleukin-6 (IL-6) secretion under the influence of Ucn-1 [42], as it has been proven that more advanced stages of EMS correspond with greater IL-6 production [43]. Although such a connection between Ucn1 and IL-6 secretion has been demonstrated in tissues not within the reproductive system, we assume that this is a general mechanism of the peptide [42].

However, not all biological properties of Ucn1 seem to be consistent with these findings obtained in our study. While more advanced forms of the EMS are inherently the result of intensified proliferation processes [44], data previously reported by others showed antiproliferative properties of Ucn1 when it is acting on endometrium [25] and on other malignant [45] and benign tissues [46].

### 3.4. Ucn1 and the Immune System in the Pathogenesis of EMS

The fact that patients with EMS present altered immune cells amounts and cytokines expression profile proves that EMS pathogenesis is largely dependent on the various immune processes [47]. The key assumptions concerning the immunological changes in patients with EMS are sometimes inconclusive and form an extensive network of dependencies that are difficult to interpret. Nevertheless, most often in the context of the immune-background of EMS, the increased number of macrophages and lymphocytes B, greater secretion of some cytokines (including IFN-gamma, IL-4 and IL-10) as well as a predominance of Th-2 cellular response and reduction in NK cells’ cytotoxic activity are mentioned [47,48].

Our study implies that the links between Ucn1 serum concentrations and various populations of immune cells, as well as the cytokines they produce, are fairly poor (Figure 6). Among patients with EMS, the presence of lymphocytes B CD19+ was positively associated with Ucn1 serum level. Additionally, when analyzing the elevated levels of peritoneal fluid Ucn1 concentrations, increased levels of lymphocytes B were observed. So far, it has been proven that Ucn1 originates from a whole panel of immune cells, including lymphocytes B and T, macrophages, monocytes and mast cells [21,49]. Thus, these findings, together with the fact that these cells share their participation in the pathogenesis of endometriosis, makes our results surprising. To our best knowledge, so far, only one research study has been focused on associations between Ucn1 and immune cells in patients with EMS. Kempuraj et al. proved that endometriotic lesions were characterized by a greater amount of mast cells and Ucn1 concentration in comparison to the normal endometrium of healthy women, hence suggesting the presence of their mutual relationships [30].

In our study, the analysis of the correlations regarding Ucn1 serum concentrations and cytokines levels including IL-2, IL-4, IL-6, IL-10 and IFN-gamma also brought not so many significant dependencies among women with endometriosis and healthy controls. We found that the patients with EMS tended to have lower peritoneal fluid IFN-gamma concentrations under conditions of elevated serum Ucn1 levels. Given the pro-inflammatory properties of IFN-gamma, our observations may support the anti-inflammatory values of Ucn1 [50]. Although, in the literature the connections between Ucn1 and IFN-gamma have not been analyzed in the group of EMS patients, the available results seem to be slightly different. Thus, the higher Ucn1 concentrations were linked with increased IFN-gamma secretion that would indicate rather the pro-inflammatory properties of Ucn1 [51,52]. Contrary to our results, in the study conducted by Novembri et al., the Ucn2 and Ucn3 were able to modify the IL-4 expression in the cultured human endometrial stromal cells (HESCs) [53]. Although the authors investigated the analogues of Ucn1, taking into consideration their partial interaction with the same receptor (CRF-R2) [54], further analysis of Ucn1 in this regard is needed.

## 4. Materials and Methods

### 4.1. Patients and Controls

A total of 76 female patients undergoing laparoscopic surgery at the 1st Department of Oncological Gynecology and Gynecology of the Medical University of Lublin, due to the EMS suspicion, were enrolled in the study group (EMS group). The control group comprised 20 patients, with no suspicion of EMS who underwent LPS, in the same hospital in order to perform a tubal patency test. The confirmation of EMS presence in the study group was made by histopathological examinations (Figure 7). Additionally, in the study group, the severity of EMS was estimated during LPS according to the revised American Society for Reproductive Medicine (rASRM) criteria.

All women participating in the study ranged between 18 and 55 years. The exclusion criteria applied to all patients were the following: co-occurrence of uterine myoma, adenomyosis, auto-immune disorders, pregnancy, lactation, allergies, as well as auto-immune and immune disorders. The medical interview reporting on past blood transfusion, oncology treatment, receiving hormonal treatment, or any symptoms of the infection within 4 weeks prior to study enrollment has also disqualified patients from the participation. Both patients and controls prior enrolling in the study provided written informed consent. 

The Institutional Review Board of the Medical University of Lublin gave ethical permission for the study protocol (No. KE-0254/302/2014). The study was conducted in adherence to the Helsinki declaration.

### 4.2. Material Collection

The day before the planned LPS, 15 mL of peripheral blood samples from the basilic vein was collected from all participants (5 mL for serum and 10 mL for EDTA anticoagulant tubes) (EDTA, Sarstedt, Germany). It was confirmed that at the time of blood collection, all patients were fasting and did not take anti-inflammatory drugs 12 and 24 h prior to blood sampling, respectively. In addition, in the study group, 5 mL of peritoneal fluid (PF) was collected immediately after the LPS procedure and then placed in EDTA-coated tubes (EDTA, Sarstedt, Germany).

### 4.3. Immunophenotyping

Flow cytometry was used to determine the immunophenotype of peripheral blood lymphocytes and the percentage of lymphocytes expressing CD25 and CD69 on their surface. The whole blood sample was stained for 20 min in the dark with the following monoclonal antibodies: anti-CD4 BV421, clone SK3 (BD Biosciences, San Jose, CA, USA), anti-CD3 PerCp, clone SP34-2 (BD Biosciences, San Jose, CA, USA), anti-CD8 BV605, clone SK1 (Biolegend, San Diego, CA, USA), anti-CD19 FITC, clone SJ25C1 (BD Biosciences, San Jose, CA, USA), anti-CD45 Alexa Fluor 700, clone 2D1 (Biolegend, San Diego, CA, USA), PE anti-human CD25, clone BC96 (Biolegend, San Diego, CA, USA), PE anti-human CD69, clone FN50 (Biolegend, San Diego, CA, USA). After incubation with antibody, the samples were treated with a lysing solution (Lysing Buffer, BD Pharm Lyse San Jose, CA, USA) prepared according to the manufacture instructions in an amount of 2 mL, and the samples were incubated for 10 min in the dark. After the incubation step with the lysis solution, the samples were centrifuged (700× *g*, 5 min) and then washed twice with a PBS solution (Sigma-Aldrich, Saint Louis, MO, USA). All samples were read on a Cytoflex LX (BeckmanCoulter, South Pasadena, CA, USA) and analyzed using the Kaluza Analysis program. The efficiency of the device and the quality control of the measuring device were carried out using CytoFLEX Ready to Use Daily QC Fluorospheres (BeckmanCoulter, CA, USA) and CytoFLEX Daily IR QC Fluorospheres (BeckmanCoulter, CA, USA) according to the manufacturers’ instructions.

### 4.4. Measurements of Serum Cytokines (IL-2, IL-4, IL-6, IL-10, IFN-gamma) Concentrations

The measurement of serum cytokines concentrations was determined with the usage of a quantitative commercial enzyme-linked immunosorbent assay (ELISA) technique, following the manufacturers’ recommendations. In brief, the serum concentration of IL-2 was measured using a Human IL-2 Quantikine ELISA Kit (sensitivity, 7 pg/mL; R&D Systems, Minneapolis, MN, USA); the serum concentration of IL-4 was determined using the Human IL-4 Quantikine HS ELISA Kit (sensitivity, 0.22 pg/mL; R&D Systems, USA); the concentration of serum IL-6 was measured using the Human IL-6 Quantikine HS ELISA Kit (sensitivity, 0.11 pg/mL; R&D Systems, USA); the serum concentration of IL-10 was determined using the Human IL-10 Quantikine HS ELISA Kit (sensitivity, 0.17 pg/mL; R&D Systems, USA), and the concentration of serum IFN-gamma was determined using the Human IFN-γ Quantikine ELISA Kit (sensitivity, 8 pg/mL; R&D Systems, USA). The results were measured with an automatic reader VICTOR3 (Perkin Elmer, Waltham, MA, USA); that action is based on the measurements of the light absorbance of the tested material and its comparison with control samples of known concentration. The WorkOut computer program, working with the reader on the basis of known concentrations, plotted linear curves on the basis of which the concentration of cytokines in the tested samples was calculated.

### 4.5. Measurement of Urocortin 1 (Ucn1) Concentration

The serum and peritoneal fluid Ucn1 concentrations were measured with an enzyme-linked immunosorbent assay (ELISA) (Urocortin (human), EIA Phoenix Pharmaceuticals INC (Burlingame, CA, USA, 94010), Cat. No: EK-019-1), according to the manufacturer’s instructions. This assay presented full compliance (100%) equivalent to urocortin 1 (Ucn1) and 0% cross-reactivity with CRH, Ucn2, Ucn3, NPY, somatostatin, Gn-RH, MCH and cortistatin 14. All samples were assessed in duplicate for each one. The detection limit of the assay was 0.2–3.8 ng/mL. The intra-assay and inter-assay coefficients of variation were less than 5% and 14%, respectively. The light absorbance was measured using the VICTOR3 automatic reader (Perkin Elmer, Boston, MA, USA).

### 4.6. Measurement of Cancer Antigen 125 (CA 125) and Human Epididymis Protein 4 (HE-4) Levels

Analysis of cancer antigen 125 (CA 125) and Human Epididymis Protein 4 (HE4) levels was performed in a Central Laboratory of Independent Public Teaching Hospital No. 1 in Lublin. Electrochemiluminescence immunoassay (ECLIA) was performed to measure the plasma concentrations of CA 125 and HE-4 in detailed via ROCHE Cobas E601 system (Roche Diagnostics GmbH, Mannheim, Germany) and the CA 125 II kit or Elecsys HE4 kit (Roche Laboratories), respectively. According to the manufacturers’ recommendations, the cutoffs for CA 125 and HE4 were 35 U/mL and 70 pmol/L, respectively.

### 4.7. Statistical Analysis

The obtained data were analyzed statistically. The significance level of statistical tests in the present analysis was set at α = 0.05. Distribution measures of the central tendency/dispersion for numerical variables were expressed in terms of Mdn (Q1–Q3). Distribution measures of central tendency/dispersion for dichotomous/ordinal/nominal variables were expressed in terms of frequencies and percentages, n (%). Estimation of mean differences between two independent groups was performed using the Mann–Whitney U test. The effect size measure was estimated using the rank biserial correlation coefficient (r_rb_). The determination and evaluation of optimal cutpoints for serum urocortin concentration, CA 125, and HE4 between the EMS and control groups were performed using the maximization metric (Sensitivity + Specificity) method with more or equal (“≥”) direction. Bootstrapping was used to evaluate the cutpoint determination methods. To measure the association between two continuous variables, the Spearman method was applied and Spearman’s rho statistic (ρ) was used to estimate a rank-based measure of association. 

Analyses were conducted using the R Statistical language (version 4.1.1; R Core Team, 2021) on a Windows 10 Pro 64 bit (build 19044). 

## 5. Conclusions

These studies have shown that Ucn1 is involved in the pathogenesis of endometriosis and significantly interacts with selected subpopulations of immune system cells and cytokine profile. Furthermore, we observed that higher serum Ucn1 levels were positively associated with the presence of DIE and more advanced disease stages. Due to the small research group diagnosed with the DIE subtype of endometriosis, it seems important to conduct further research, allowing for a clear statement of whether Ucn1 can be used in the future as one of the non-invasive biomarkers supporting the diagnostic process and indicating the severity of the disease and the risk of developing DIE.

## Figures and Tables

**Figure 1 ijms-24-07787-f001:**
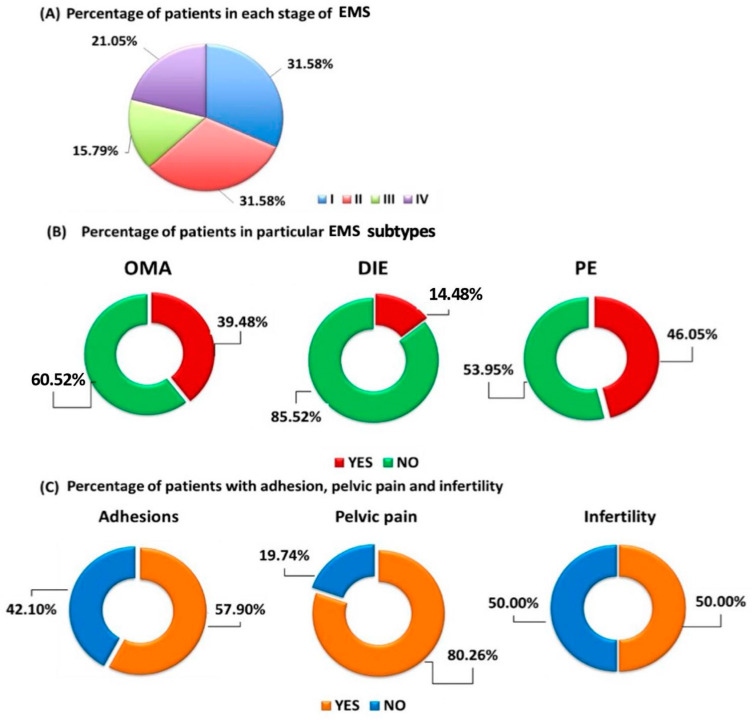
Characteristics of selected parameters of EMS patients. (**A**) Differentiation of EMS patients by stage; (**B**) Number of patients in particular EMS subtypes; (**C**) Selected symptoms of EMS patients.

**Figure 2 ijms-24-07787-f002:**
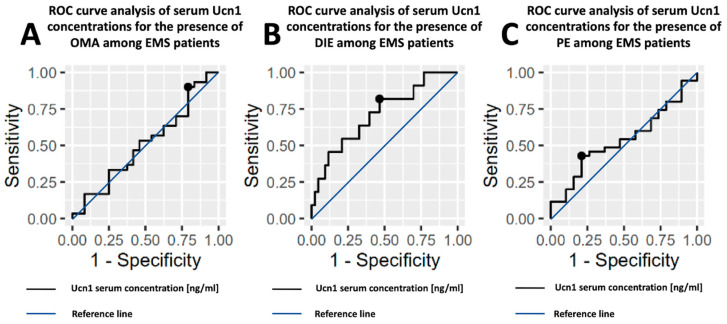
Receiver operating characteristic (ROC) curve analysis showing the sensitivity and specificity of Ucn1 serum concentration in different EMS types: (**A**)—in OMA, (**B**)—in DIE and (**C**)—in PE. Black points refer to cutoff values.

**Figure 3 ijms-24-07787-f003:**
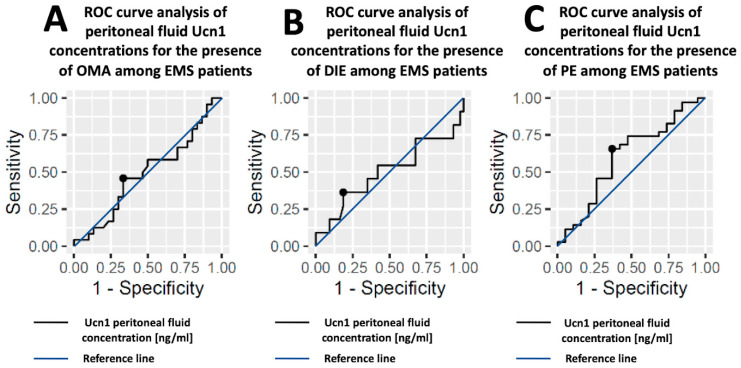
Receiver operating characteristic (ROC) curve analysis showing the sensitivity and specificity of Ucn1 concentration in peritoneal fluid in different EMS types: (**A**)—in OMA, (**B**)—in DIE and (**C**)—in PE. Black points refer to cutoff values.

**Figure 4 ijms-24-07787-f004:**
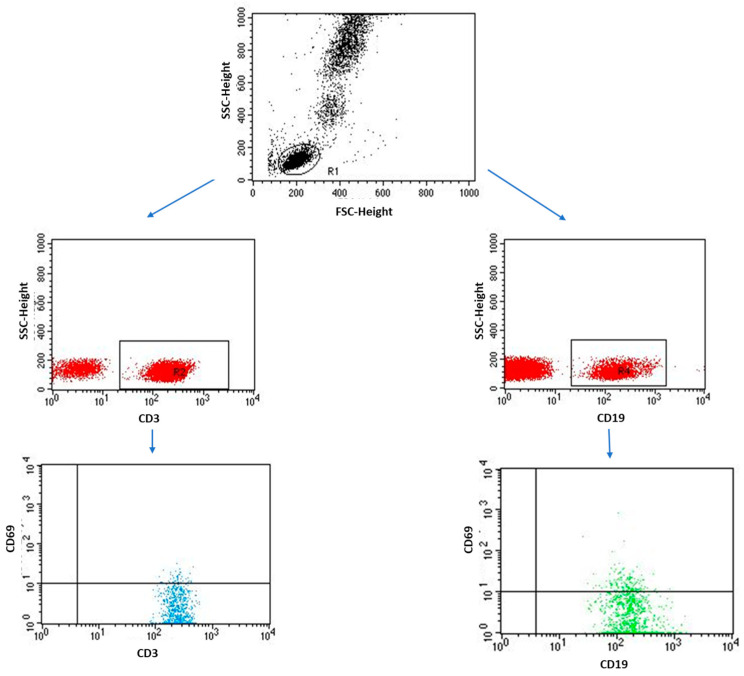
Sample analysis of the percentage of lymphocytes with the CD3+/CD69+ and CD19+/CD69+ phenotype.

**Figure 5 ijms-24-07787-f005:**
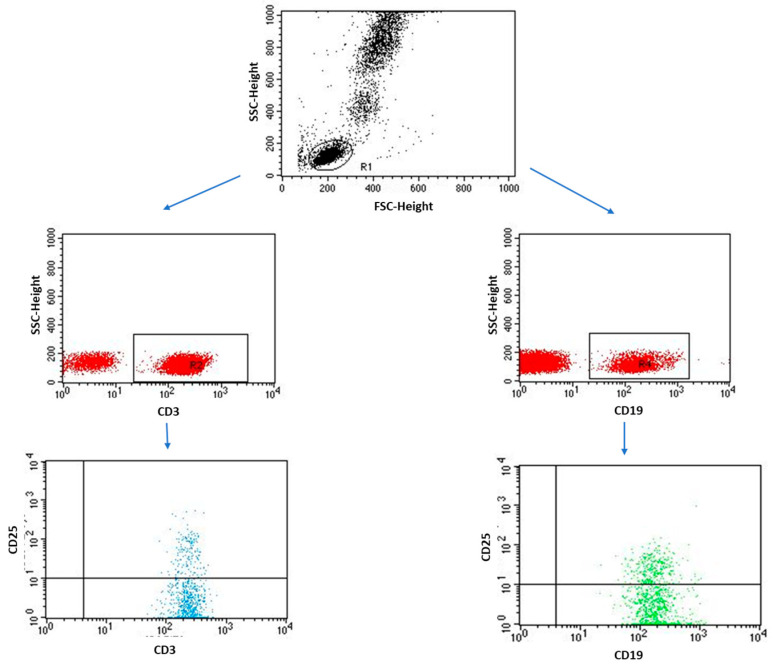
Sample analysis of the percentage of lymphocytes with the CD3+/CD25+ and CD19+/CD25+ phenotype.

**Figure 6 ijms-24-07787-f006:**
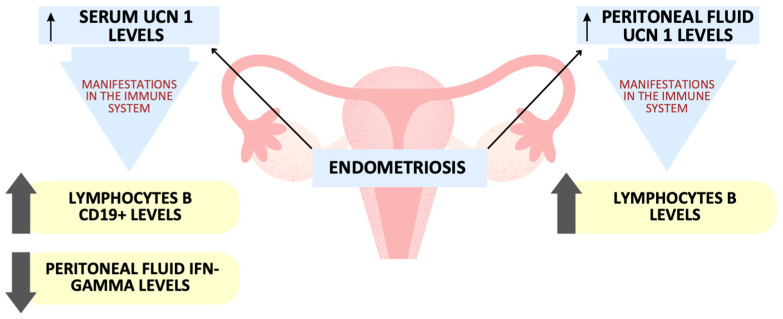
Observed associations between high urocortin 1 concentrations and immune system components.

**Figure 7 ijms-24-07787-f007:**
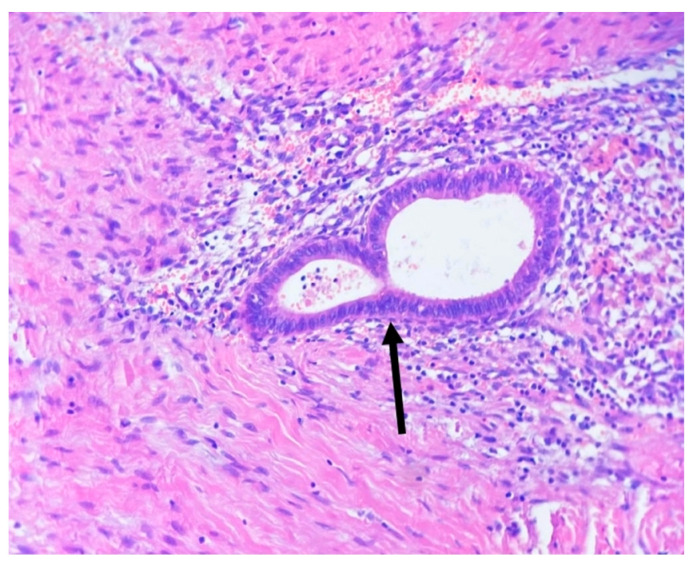
Picture of endometrial lesion in peritoneum (stained with H+E; 10× magnification). Arrow indicates endometrioid gland. Around the endometrioid gland, the immune cell infiltration is visible.

**Table 1 ijms-24-07787-t001:** Characteristics of selected parameters of peripheral blood morphology and biochemistry of patients from the study group and the control group.

Parameter	Patients with EMS (n = 76)		Control Group (n = 20)		*p*-Value
	Mean ± SD	Median (Range)	Mean ± SD	Median (Range)	
**Age**	35.00 ± 6.14	35.00 (22.00–48.00)	36.65 ± 6.69	36.00 (26.00–53.00)	0.428
White blood cells (10^3^ /mm^3^)	8.39 ± 1.59	8.31 (5.51–11.33)	7.42 ± 0.75	7.31 (6.37–8.66)	0.014 *
Neutrophils (10^3^ /mm^3^)	5.04 ± 1.38	5.50 (2.08–7.91)	4.32 ± 1.01	3.93 (2.71–6.03)	0.0138 *
Monocytes (10^3^ /mm^3^)	0.54 ± 0.16	0.56 (0.24–0.86)	0.46 ± 0.09	0.48 (0.28–0.59)	0.053
Lymphocytes (10^3^ /mm^3^)	2.24 ± 0.72	2.05 (1.20–3.83)	2.43 ± 0.44	2.53 (1.53–3.07)	0.188
TSH (thyrotrophic hormone) (μIU/mL)	1.38 ± 0.62	1.40 (0.33–2.58)	1.42 ± 0.67	1.49 (0.44–2.55)	0.813
fT3 (free triiodothyronine) (pg/mL)	3.15 ± 0.56	3.15 (2.01–4.34)	3.13 ± 0.56	3.15 (2.02–4.12)	0.870
fT4 (free thyroxin) (pg/mL)	1.31 ± 0.24	1.32 (0.92–1.73)	1.29 ± 0.21	1.30 (0.93–1.67)	0.758
Estradiol (pg/mL)	57.93 ± 26.18	51.00 (19.30–114.80)	56.23 ± 26.74	44.95 (25.6–112.30)	0.659
FSH (follicle-stimulating hormone) (mIU/mL)	5.36 ± 1.79	4.90 (2.20–9.50)	6.16 ± 1.54	6.00 (3.50–8.50)	0.068
LH (luteinizing hormone) (mIU/mL)	6.64 ± 2.73	6.00 (2.20–14.60)	5.96 ± 1.78	5.75 (3.20–10.50)	0.525

* Statistically significant results.

**Table 2 ijms-24-07787-t002:** Evaluation of Ucn1, CA 125 and HE4 concentrations among patients from the research and control groups.

Parameter	Patients with EMS (n = 76)		Control Group (n = 20)		*p*-Value
	Mean ± SD	Median (Range)	Mean ± SD	Median (Range)	
Level of Ucn1 in serum (ng/mL)	12.38 ± 6.18	11.78 (1.75–26.85)	4.52 ± 2.71	3.50 (1.30–10.55)	0.0000 *
Level Ucn1 in peritoneal fluid (ng/mL)	1.30 ± 0.68	1.22 (0.15–2.52)	N/A	N/A	N/A
Level of CA 125 (U/mL)	37.78 ± 28.68	30.84 (7.67–178.12)	9.23 ± 5.23	9.02 (2.50–21.73)	0.0000 *
Level of HE4 (pmol/L)	39.96 ± 9.60	39.15 (26.00–67.00)	37.05 ± 8.90	37.63 (16.92–53.18)	0.414

* Statistically significant results; N/A—not applicable.

**Table 3 ijms-24-07787-t003:** Evaluation of the effectiveness of Ucn1, CA 125 and HE4 as potential biomarker molecules of EMS development.

Parameter	Optimal Cutpoint	AUC	N	N Positive	N Negative	TP	FN	FP	TN	Se	Sp	Sp + Se	Acc
Serum Ucn1 (ng/mL)	5.94	0.85	96	76	20	58	11	5	21	0.83	0.80	1.63	0.82
CA 125 (U/mL)	16.87	0.94	96	76	20	58	11	2	25	0.83	0.95	1.78	0.87
HE4 (pmol/L)	40.4	0.56	96	76	20	30	40	7	19	0.42	0.75	1.18	0.51

Abbreviations: Ucn1—urocortin 1; CA 125—cancer antigen-125; HE4—human epididymis protein; N—the sample size; TP—vector of true positives; FN—vector of false negatives; FP—vector of false positives; TN—vector of true negatives; Se—sensitivity; Sp—specificity; Acc—accuracy; N positive = TP+FN; N negative = FP+TN; Se = TP/(TP+FN); Sp = TN/(TN+FP); Acc = (TP+TN)/(TP+FP+TN+FN).

**Table 4 ijms-24-07787-t004:** Evaluation of Ucn1, CA 125 and HE4 concentrations between individual EMS subtypes and the control group.

Parameter	OMA (n = 30) (Group 1)		PE (n = 35) (Group 2)		DIE (n = 11) (Group 3)		Control (n = 20) (Group 4)		*p*-Value						
	Mean ±SD	Median (Range)	Mean ±SD	Median (Range)	Mean ±SD	Median (Range)	Mean ±SD	Median (Range)	*p*-Value	1 vs. 2	1 vs. 3	1 vs. 4	2 vs. 3	2 vs. 4	3 vs. 4
Level of Ucn1 in serum	12.37 ± 6.49	11.87 (2.43–26.85)	12.83 ± 7.20	11.86 (1.75–26.85)	16.70 ± 6.86	16.20 (6.32–26.85)	4.52 ± 2.71	3.50 (1.30–10.55)	0.000 *	0.911	0.103	0.000 *	0.144	0.000 *	0.000 *
Level of CA 125	44.12 ± 32.64	33.47 (9.98–178.12)	33.98 ± 23.43	29.26 (7.67–101.00)	54.85 ± 45.11	32.56 (17.11–178.12)	9.23 ± 5.23	9.02 (2.50–21.73)	0.000 *	0.129	0.551	0.000 *	0.089	0.000 *	0.000 *
Level of HE4	41.81 ± 10.85	40.25 (26.00–69.00)	38.61 ± 8.99	38.20 (26.00–67.00)	43.65 ± 7.34	42.30 (31.00–59.30)	37.05 ± 8.90	37.63 (16.92–53.18)	0.136	0.268	0.374	0.000 *	0.033 *	0.775	0.014 *

* Statistically significant results.

**Table 5 ijms-24-07787-t005:** Evaluation of selected subpopulations of T and B lymphocytes affecting the functioning of the immune system of patients diagnosed with EMS and the control group.

Parameter		Patients with EMS (n = 76)		Control Group (n = 20)		*p*-Value
		Mean ± SD	Median (Range)	Mean ± SD	Median (Range)	
Frequency of occurrence of individual subpopulations of immune cells in peripheral blood (%)	CD3+ T lymphocytes	70.83 ± 4.59	71.86 (61.31–78.77)	68.26 ± 3.74	68.08 (60.63–74.49)	0.020 *
	CD3+/CD4+ T lymphocytes	43.09 ± 7.56	44.01 (26.13–65.45)	44.46 ± 2.44	44.16 (40.71–48.84)	0.731
	CD3+/CD8+ T lymphocytes	28.07 ± 6.79	27.99 (16.25–42.50)	34.36 ± 3.20	34.74 (29.33–39.60)	0.0000 *
	CD4+/CD25/Foxp3+ T lymphocytes	6.30 ± 3.13	5.46 (0.39–13.55)	6.20 ± 1.69	6.25 (3.13–9.68)	0.956
	CD19+ B lymphocytes	10.55 ± 3.08	9.76 (6.12–16.84)	11.25 ± 2.44	11.40 (6.04–16.90)	0.176
	CD3+/CD25+ T lymphocytes	28.58 ± 8.32	26.96 (10.86–56.29)	7.60 ± 2.69	8.03 (1.08–11.13)	0.0000 *
	CD4+/CD25+ T lymphocytes	13.90 ± 6.03	14.36 (0.82–29.48)	5.66 ± 2.40	6.35 (0.95–8.83)	0.0000 *
	CD8+/CD25+ T lymphocytes	14.69 ± 8.83	13.66 (2.33–34.98)	1.94 ± 1.11	1.63 (0.13–5.11)	0.0000 *
	CD19+/CD25+ B lymphocytes	3.50 ± 1.74	3.10 (0.55–8.14)	1.77 ± 1.29	1.81 (0.06–5.12)	0.0001 *
	CD3+/CD69+ T lymphocytes	14.80 ± 8.69	14.65 (1.21–33.66)	3.38 ± 1.66	3.36 (0.52–6.89)	0.0000 *
	CD4+/CD69+ T lymphocytes	9.52 ± 5.58	9.42 (0.78–21.64)	2.20 ± 1.00	2.30 (0.18–3.48)	0.0000 *
	CD8+/CD69+ T lymphocytes	5.29 ± 3.10	5.23 (0.43–12.02)	1.18 ± 1.19	0.70 (0.02–3.87)	0.0000 *
	CD19+/CD69+ B lymphocytes	2.17 ± 0.95	2.02 (0.06–6.39)	0.12 ± 0.06	0.09 (0.06–0.25)	0.0000 *

* Statistically significant results.

**Table 6 ijms-24-07787-t006:** Evaluation of selected subpopulations of T and B lymphocytes affecting the functioning of the patient’s immune system between individual EMS subtypes and the control group.

Frequency of Occurrence of Individual Subpopulations of Immune Cells in Peripheral Blood	OMA (n = 30) (Group 1)		PE (n = 35) (Group 2)		DIE (n = 11) (Group 3)		Control (n = 20) (Group 4)		*p*-Value						
	Mean ± SD	Median (Range)	Mean ± SD	Median (Range)	Mean ± SD	Median (Range)	Mean ± SD	Median (Range)	*p*-Value	1 vs. 2	1 vs. 3	1 vs. 4	2 vs. 3	2 vs. 4	3 vs. 4
CD3+ T lymphocytes	70.32 ± 4.29	69.94 (61.88–76.55)	70.76 ± 4.93	72.08 (61.31–78.77)	72.33 ± 5.60	74.40 (61.31–78.77)	68.26 ± 3.74	68.08 (60.63–74.49)	0.074	0.615	0.183	0.087 *	0.372	0.042 *	0.054
CD3+/CD4+ T lymphocytes	41.97 ± 8.00	43.78 (26.13–54.97)	41.81 ± 6.20	43.20 (26.62–53.10)	45.27 ± 9.60	46.59 (27.34–65.45)	44.46 ± 2.44	44.16 (40.71–48.84)	0.525	0.605	0.441	0.536	0.320	0.182	0.982
CD3+/CD8+ T lymphocytes	26.85 ± 6.78	25.35 (16.37–42.90)	29.31 ± 6.45	29.06 (18.13–42.90)	23.88 ± 7.93	18.73 (16.25–40.97)	34.36 ± 3.20	34.74 (29.33–39.60)	0.000 *	0.086	0.164	0.000 *	0.027 *	0.000 *	0.001 *
CD4+/CD25/Foxp3+ T lymphocytes	5.74 ± 3.10	4.57 (1.09–13.55)	6.39 ± 2.97	6.46 (0.39–13.55)	9.88 ± 2.66	10.21 (3.58–13.55)	6.20 ± 1.69	6.25 (3.13–9.68)	0.003 *	0.303	0.000 *	0.350	0.001 *	0.896	0.001 *
CD19+ B lymphocytes	10.41 ± 3.19	9.76 (6.12–16.83)	10.80 ± 3.04	10.64 (6.12–16.84)	11.53 ± 3.09	11.40 (7.69–16.82)	11.25 ± 2.44	11.40 (6.04–16.90)	0.509	0.569	0.329	0.164	0.558	0.380	0.812
CD3+/CD25+ T lymphocytes	26.93 ± 7.52	25.96 (10.86–41.20)	29.03 ± 8.63	27.03 (10.86–56.29)	34.63 ± 9.84	29.99 (23.37–56.29)	7.60 ± 2.69	8.03 (1.08–11.13)	0.000 *	0.406	0.036 *	0.000 *	0.117	0.000 *	0.000 *
CD4+/CD25+ T lymphocytes	13.05 ± 6.03	13.5 (0.82–29.48)	13.95 ± 5.46	14.40 (0.82–22.10)	18.96 ± 3.53	18.20 (11.08–25.90)	5.66 ± 2.40	6.35 (0.95–8.83)	0.000 *	0.362	0.001 *	0.000 *	0.004 *	0.000 *	0.000 *
CD8+/CD25+ T lymphocytes	13.88 ± 8.19	13.68 (2.75– 34.05)	15.08 ± 9.64	12.04 (2.23–34.08)	15.68 ± 8.70	12.29 (5.29–34.98)	1.94 ± 1.11	1.63 (0.13–5.11)	0.000 *	0.624	0.532	0.000 *	0.741	0.000 *	0.000 *
CD19+/CD25+ B lymphocytes	3.32 ± 1.77	2.94 (0.55–8.14)	3.41 ± 1.43	3.10 (1.35–7.33)	5.52 ± 1.89	5.93 (1.99–8.14)	1.77 ± 1.29	1.81 (0.06–5.12)	0.000 *	0.596	0.003 *	0.000 *	0.003*	0.000 *	0.000 *
CD3+/CD69+ T lymphocytes	13.82 ± 7.55	13.68 (1.24–27.30)	14.09 ± 9.26	12.75 (1.21–33.66)	7.14 ± 7.02	4.41 (1.21–21.11)	3.38 ± 1.66	3.36 (0.52–6.89)	0.000 *	1.00	0.027 *	0.000 *	0.025 *	0.000 *	0.619
CD4+/CD69+ T lymphocytes	8.88 ± 4.85	8.80 (0.80–17.55)	9.06 ± 5.95	8.20 (0.78–21.64)	4.59 ± 4.51	2.84 (0.78–13.57)	2.20 ± 1.00	2.30 (0.18–3.48)	0.000 *	1.00	0.027 *	0.000 *	0.025 *	0.000 *	0.619
CD8+/CD69+ T lymphocytes	4.94 ± 2.70	4.89 (0.44–9.75)	5.03 ± 3.31	4.55 (0.43–12.02)	2.55 ± 2.51	1.58 (0.43–7.54)	1.18 ± 1.19	0.70 (0.02–3.87)	0.000 *	0.994	0.021 *	0.000 *	0.003 *	0.000 *	0.074
CD19+/CD69+ B lymphocytes	2.08 ± 0.91	2.00 (0.82–6.39)	2.15 ± 0.88	1.99 (0.06–4.59)	1.98 ± 0.86	2.10 (0.06–3.25)	0.12 ± 0.06	0.09 (0.06–0.25)	0.000 *	0.680	0.424	0.000 *	0.899	0.000 *	0.000 *

* Statistically significant results.

**Table 7 ijms-24-07787-t007:** Quantitative assessment of the concentration of cytokines IL-2, IL-4, IL-6, IL-10, and IFN-γ in the test material from patients with EMS in relation to patients from the control group.

Parameter	Patients with EMS (n = 76)		Control Group (n = 20)		*p*-Value
	Mean ± SD	Median (Range)	Mean ± SD	Median (Range)	
IL-4 concentration in serum (pg/mL)	3.14 ± 2.22	2.77 (0.47–12.04)	4.80 ± 0.31	4.67 (4.29–5.51)	0.000 *
IL-6 concentration in serum (pg/mL)	22.00 ± 7.12	22.06 (5.17–41.38)	3.11 ± 4.03	1.17 (0.15–17.20)	0.000 *
IL-10 concentration in serum (pg/mL)	10.22 ± 12.22	6.77 (0.07–65.30)	3.96 ± 0.97	3.91 (2.77–6.16)	0.013 *
IL-2 concentration in serum (pg/mL)	14.20 ± 12.89	8.18 (0.98–50.12)	2.76 ± 1.95	2.36 (0.48–7.16)	0.000 *
IFN-γ concentration in serum (pg/mL)	6.45 ± 3.75	5.45 (1.28–23.30)	2.33 ± 1.16	2.34 (0.61–4.38)	0.000 *
IL-4 concentration in peritoneal fluid (pg/mL)	3.34 ± 2.93	2.32 (0.42–14.93)	N/C	N/C	N/A
IL-6 concentration in peritoneal fluid (pg/mL)	141.02 ± 29.57	66.72 (10.24–395.58)	N/C	N/C	N/A
IL-10 concentration in peritoneal fluid (pg/mL)	41.41 ± 11.30	33.58 (10.04–156.88)	N/C	N/C	N/A
IL-2 concentration in peritoneal fluid (pg/mL)	17.15 ± 4.44	16.80 (1.19–27.11)	N/C	N/C	N/A
IFN-γ concentration in peritoneal fluid (pg/mL)	112.34 ± 37.19	99.21 (13.34–496.38)	N/C	N/C	N/A

* Statistically significant results; N/C—peritoneal fluid was not collected from the control participants; N/A—not applicable.

**Table 8 ijms-24-07787-t008:** Quantitative assessment of the concentration of cytokines IL-2, IL-4, IL-6, IL-10, and IFN-γ in the examined material depending on the EMS subtype in relation to patients from the control group.

Parameter	OMA (n = 30) (Group 1)		PE (n = 35) (Group 2)		DIE (n = 11) (Group 3)		Control (n = 20) (Group 4)		*p*-Value						
	Mean ± SD	Median (Range)	Mean ± SD	Median (Range)	Mean ± SD	Median (Range)	Mean ± SD	Median (Range)	*p*-Value	1 vs. 2	1 vs. 3	1 vs. 4	2 vs. 3	2 vs. 4	3 vs. 4
IL-4 concentration in serum (pg/mL)	3.28 ± 2.52	2.77 (0.64–12.04)	3.40 ± 2.48	2.97 (0.47–12.04)	2.38 ± 1.41	2.54 (0.86–6.20)	4.80 ± 0.31	4.67 (4.29–5.51)	0.001 *	0.683	0.249	0.000 *	0.158	0.002 *	0.003 *
IL-6 concentration in serum (pg/mL)	30.59 ± 8.07	22.28 (0.44–41.38)	18.93 ± 8.23	11.98 (0.17–38.98)	40.57 ± 10.48	31.98 (0.46–37.98)	3.11 ± 4.03	1.17 (0.15–17.20)	0.001 *	0.029 *	0.076	0.180	0.001 *	0.000 *	0.000 *
IL-10 concentration in serum (pg/mL)	12.64 ± 14.34	9.09 (0.07–65.30)	9.66 ± 13.84	5.71 (0.07–65.30)	13.69 ± 7.49	9.48 (1.18–65.30)	3.96 ± 0.97	3.91 (2.77–6.16)	0.027 *	0.059 *	0.94	0.002 *	0.035 *	0.000 *	0.000 *
IL-2 concentration in serum (pg/mL)	12.42 ± 6.01	7.33 (0.98–50.12)	17.41 ± 7.16	9.60 (1.29–50.12)	26.07 ± 10.84	19.62 (1.29–50.12)	2.76 ± 1.95	2.36 (0.48–7.16)	0.000 *	0.042 *	0.001 *	0.000 *	0.027 *	0.000 *	0.000 *
IFN-γ concentration in serum (pg/mL)	6.43 ± 3.47	5.56 (3.83–23.30)	6.48 ± 4.28	5.44 (1.28–23.30)	6.65 ± 2.88	5.29 (4.75–13.40)	2.33 ± 1.16	2.34 (0.61–4.38)	0.000 *	0.633	0.980	0.000 *	0.780	0.000 *	0.000 *
IL-4 concentration in peritoneal fluid (pg/mL)	3.54 ± 3.27	2.32 (0.42–14.93)	2.92 ± 2.28	2.02 (0.42–10.29)	5.65 ± 4.06	5.92 (0.71–14.93)	N/C	N/C	N/A	0.690	0.024 *	N/A	0.001 *	N/A	N/A
IL-6 concentration in peritoneal fluid (pg/mL)	112.21 ± 30.43	64.40 (10.39–395.56)	166.52 ± 37.81	166.77 (50.24–387.91)	49.84 ± 22.51	37.80 (10.24–44.16)	N/C	N/C	N/A	0.000 *	0.032 *	N/A	0.00 *	N/A	N/A
IL-10 concentration in peritoneal fluid (pg/mL)	29.89 ± 11.36	21.08 (10.04–99.63)	48.26 ± 7.24	23.51 (10.04–56.88)	17.73 ± 7.78	11.17 (3.63–65.71)	N/C	N/C	N/A	0.480	0.001 *	N/A	0.000 *	N/A	N/A
IL-2 concentration in peritoneal fluid (pg/mL)	20.94 ± 7.99	6.60 (1.19–27.11)	13.08 ± 7.53	6.31 (1.19–26.80)	8.04 ± 3.53	8.58 (2.78–22.64)	N/C	N/C	N/A	0.279	0.059	N/A	0.062	N/A	N/A
IFN-γ concentration in peritoneal fluid (pg/mL)	139.85 ± 51.75	111.73 (13.34–496.38)	123.54 ± 57.77	106.73 (33.34–495.29)	91.91 ± 20.04	85.86 (23.86–380.88)	N/C	N/C	N/A	0.436	0.001 *	N/A	0.037 *	N/A	N/A

* Statistically significant results; N/C—peritoneal fluid was not collected from the control participants; N/A—not applicable.

## Data Availability

The data presented in this study are available on request from the corresponding author.

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
