# Peer review of "Possible Correlation between Urocortin 1 (Ucn1) and Immune Parameters in Patients with Endometriosis"

_ijms, 2023, doi:10.3390/ijms24097787_

Round 1
Reviewer 1 Report
In this paper by Abramiuk and coworkers, the authors investigate the role of urocortin 1 in the context of endometriosis, hypothesizing that this molecule could serve as a biomarker of the disease. To me it seems from the literature mentioned that it plays the role of an anti-inflammatory peptide; can the author confirm?
The authors studied 76 patients and a control group of 20 women. The study on the blood is quite exhaustive and provides original insights into the immune regulation in endometriotic women, that may not have been studied so extensively previously. The authors first report significant differences in the blood composition between the two groups for neutrophils and WBC taken as a whole. In the serum, the concentration of UCN1 was considerably increased in the patients, with an AUC in ROC curve of .85 almost as good as CA125. UCN1 was not able to distinguish OMA inside the patients, but displayed some efficacy in diagnosing DIE compared to other endometriosis cases. UCN1 was somehow redundant with CA125 in separating the different types of endometriosis.
Interesting complementary data on subpopulations of lymphocytes, revealed significant differences between patients and controls for specific subpopulations of CD3+ T lymphocytes and CD19+ B lymphocytes. Several cytokines were also altered.
Major criticisms:
Paradoxically probably the part on urocortin appears a bit secondary in the paper, despite the title, and its usefulness to segregate different endometriosis types is still doubtful. There is no specific demonstration of the role of urocortin in modifying the blood formula, and this appears rather as a parallel observation. As such the conclusion that UCN1 interacts with subpopulation of immune cells seem unsubstantiated. There is a correlation, but I did not see the proof of causality at the molecular or cellular level.
As such I feel that the causality in the title should disappear. Another title corresponding to the results could be something like ‘Altered blood formula and parameters in endometriotic patients; potential correlation with circulating urocortin levels’.
Throughout the text, I think that less emphasis should be given to Ucn1 usefulness, while as mentioned, it could be moderately interesting for discriminating DIE from the other forms of endometriosis.
Minor criticisms:
What does the author mean by saying that Ucn1 was identified in the brain acupuncture?
Line 90: written ‘ESM’ and not ‘EMS’, as well as in several places in the text.
Figures 2 and 3 are exactly the same apparently.
Author Response
Dear Reviewer,
First and foremost thank You very much for all the valuable suggestions and comments. We are aware that in its past form our manuscript suffered from several shortcomings. However, we introduced some changes in our paper following Your guidance with the hope for obtaining its improved version. We believe that now our paper will meet with Your approval. Please find the more detailed responds to Your suggestions listed below.
- In this paper by Abramiuk and coworkers, the authors investigate the role of urocortin 1 in the context of endometriosis, hypothesizing that this molecule could serve as a biomarker of the disease. To me it seems from the literature mentioned that it plays the role of an anti-inflammatory peptide; can the author confirm?
- Thank You for this question. When analyzing the literature, the reports about anti-inflammatory properties of Ucn1 can be found, thus in the “Introduction” section we have mentioned about this term. Also based on the results of our work indicating that urocortin 1 was elevated in severe forms, as well as in more advanced stages of the disease, it can be hypothesized that it was secreted in larger quantities to counteract the associated inflammation. However, as reports of role of urocortin 1 are still developing, and our study was the first showing the correlation between urocortin 1 and disease stage, it seems that urocortin 1 defined as anti-inflammatory peptide should be understood as possible conjecture rather than, as far-reaching conclusions.
- The authors studied 76 patients and a control group of 20 women. The study on the blood is quite exhaustive and provides original insights into the immune regulation in endometriotic women, that may not have been studied so extensively previously. The authors first report significant differences in the blood composition between the two groups for neutrophils and WBC taken as a whole. In the serum, the concentration of UCN1 was considerably increased in the patients, with an AUC in ROC curve of .85 almost as good as CA125. UCN1 was not able to distinguish OMA inside the patients, but displayed some efficacy in diagnosing DIE compared to other endometriosis cases. UCN1 was somehow redundant with CA125 in separating the different types of endometriosis. Interesting complementary data on subpopulations of lymphocytes, revealed significant differences between patients and controls for specific subpopulations of CD3+ T lymphocytes and CD19+ B lymphocytes. Several cytokines were also altered.
- Thank You for this comment.
- Paradoxically probably the part on urocortin appears a bit secondary in the paper, despite the title, and its usefulness to segregate different endometriosis types is still doubtful. There is no specific demonstration of the role of urocortin in modifying the blood formula, and this appears rather as a parallel observation. As such the conclusion that UCN1 interacts with subpopulation of immune cells seem unsubstantiated. There is a correlation, but I did not see the proof of causality at the molecular or cellular level.
As such I feel that the causality in the title should disappear. Another title corresponding to the results could be something like ‘Altered blood formula and parameters in endometriotic patients; potential correlation with circulating urocortin levels’.
Throughout the text, I think that less emphasis should be given to Ucn1 usefulness, while as mentioned, it could be moderately interesting for discriminating DIE from the other forms of endometriosis. - Thank You for Your opinion. We decided to change the title of our manuscript following Your suggestions. We believe that the emphasizing in the title of our paper, only the potential association of observed changes in the immune system with urocortin in patients with endometriosis can generally improve value of our manuscript. We also corrected some part of the text accordingly.
Minor criticisms:
- What does the author mean by saying that Ucn1 was identified in the brain acupuncture?
- Thank You for this question. This sentence was supposed to refer to the primary description of urocortin 1 and its connection with brain tissue. However, fearing that readers might misunderstand this fragment, we decided to reformulate it.
- Line 90: written ‘ESM’ and not ‘EMS’, as well as in several places in the text.
- Thank You for paying attention to that. The abbreviations have been corrected in all places in the text.
- Figures 2 and 3 are exactly the same apparently.
- Thank You for this comment. We have removed one of these Figures with an additional changing of the preceding text fragments.
On Behalf of the authors,
Monika Abramiuk
Reviewer 2 Report
The title of the manuscript is good but the novelity in some parts of the results of the manuscript needs some explaination. English language is in good quality. Figures need some changes.The multiple references are the citation problems in this manuscript. The authors should exert some modifications in the part "Discussion"
1. One of the main part of the results of the manuscript tries to display the role of Ucn1 in the diognosis and pathogenesis of endometriosis. Please explain what is the novelity of your work in comparison with these two manuscript below?
In fact, these two manuscript have remarked the role of Ucn1 in pathigenesis and diognosis of endometriosis. Thus, I am curious to know what is the main difference(s) of your work with remarked articles?
1) Urocortin Expression in Endometriosis: A
Systematic Review"
Vasilios Pergialiotis, M.D, Ph.D, Nikoletta Maria Tagkou, M.D, [...], and Pantelis Trompoukis, M.D, Ph.D
2) Endometriosis: New Perspective for the Diagnosis of Certain Cytokines in Women and Adolescent Girls, as Well as the Progression of Disease Outgrowth: A Systematic Review
Jakub Toczek * , Zaneta Jastrz ˛ebska-Stojko, Rafał Stojko ˙ and Agnieszka Drosdzol-Cop
2. About the part "Discussion"
Please categorize this part based on your most important result to the least important one. After that, turn each of these results into subheading. Then, discuss about them one by one.
3. Please reform all multipple references in the text of manuscript
4. Please demonstrate interactions between Ucn1and the immune system in the pathogenesis of endometriosis in the form of one simple figure (based on the results of the manuscript) in the part "Discussion"
5. The most important result of the manuscript is the interaction between Ucn1and the immune system in endometriosis. Thus, it is necessary to discuss in detail about logican connections between these two factor and endometriosis.
6. Why figure 5 and 6 are in the part "materials and methods"? Was it not better to insert these two figures on the part"Results"?
7. In page 5, line 383-385
The authors have mentioned that "The confirmation or exclusion of EMS presence (in the study and control group, respectively) was made by histopathological examinations.)
Please explain why you have not inserted any figure about mentioned histopathological examination in the manuscript?
8. Please check and adjust the "Reference list" based on the regulations of reference list of journal. (Titles, doi, the name of journal and ... )
Author Response
Dear Reviewer,
First and foremost thank You very much for all the valuable suggestions and comments. We are aware that in its past form our manuscript suffered from several shortcomings. However, we introduced some changes in our paper following Your guidance with the hope for obtaining its improved version. We believe that now our paper will meet with Your approval. Please find the more detailed responds to Your suggestions listed below.
- The title of the manuscript is good but the novelity in some parts of the results of the manuscript needs some explaination. English language is in good quality. Figures need some changes. The multiple references are the citation problems in this manuscript. The authors should exert some modifications in the part "Discussion"
- Thank You for Your opinion. We made the changes following Your suggestions to make our manuscript more valuable.
- One of the main part of the results of the manuscript tries to display the role of Ucn1 in the diagnosis and pathogenesis of endometriosis. Please explain what is the novelity of your work in comparison with these two manuscript below?
In fact, these two manuscript have remarked the role of Ucn1 in pathogenesis and diagnosis of endometriosis. Thus, I am curious to know what is the main difference(s) of your work with remarked articles?
1) Urocortin Expression in Endometriosis: A
Systematic Review"
Vasilios Pergialiotis, M.D, Ph.D, Nikoletta Maria Tagkou, M.D, [...], and Pantelis Trompoukis, M.D, Ph.D
2) Endometriosis: New Perspective for the Diagnosis of Certain Cytokines in Women and Adolescent Girls, as Well as the Progression of Disease Outgrowth: A Systematic Review
Jakub Toczek * , Zaneta Jastrzebska-Stojko, Rafał Stojko and Agnieszka Drosdzol-Cop
- Thank You for this question. Although our study to some extent shares main topic with the above-mentioned papers, there are several relevant arising issues which distinguish our manuscript. We are all in agreement about high scientific value of both systematic reviews, however we believe our paper shed a new light on the link between Ucn1 and the immune system in the patients with endometriosis. First, the systematic review by Pergialotis et al. exclusively focused on the role of urocortin 1 in endometriosis and the authors did not analyze the relationship between Ucn1 and endometriosis with the simultaneous taking into consideration the immune background of both the peptide and disease. Further, while we find the systematic review by Toczek et al. as a valuable regarding the evaluation of endometriosis markers, in our opinion the part concerning the urocortin 1 is slightly superficial.
- About the part "Discussion"
Please categorize this part based on your most important result to the least important one. After that, turn each of these results into subheading. Then, discuss about them one by one.
- Thank You for Your opinion. The part “Discussion” was corrected according to Your suggestions.
- Please reform all multipple references in the text of manuscript
- Thank You for paying attention to that. All mistakes concerning references were corrected.
- Please demonstrate interactions between Ucn1 and the immune system in the pathogenesis of endometriosis in the form of one simple figure (based on the results of the manuscript) in the part "Discussion"
- Thank You for Your suggestions. We have summarized the part of the “Results” concerning the Ucn1 and the immune system in the pathogenesis of EMS, via attached simple Figure No. 7 in the “Discussion” section.
- The most important result of the manuscript is the interaction between Ucn1 and the immune system in endometriosis. Thus, it is necessary to discuss in detail about logican connections between these two factor and endometriosis.
- Thank You for Your opinion. We have developed a paragraph in the “Discussion” regarding the association between Ucn1 and EMS.
- Why figure 5 and 6 are in the part "materials and methods"? Was it not better to insert these two figures on the part"Results"?
- Thank You for this opinion. We decided to insert Figure 5 and 6 in the “Results” section.
- In page 5, line 383-385
The authors have mentioned that "The confirmation of EMS presence (in the study and control group, respectively) was made by histopathological examinations.)
Please explain why you have not inserted any figure about mentioned histopathological examination in the manuscript?
- Thank You for this comment. According to Your suggestions, we have added a figure presenting the histopathological evaluation of endometrial lesion in peritoneum (stained with H+E; magnification 10x).
- Please check and adjust the "Reference list" based on the regulations of reference list of journal. (Titles, doi, the name of journal and ...)
- We corrected this part of our manuscript accordingly.
Again, we would like to thank the Reviewer for the effort and we are hoping that our manuscript in its current form will fulfill the requirements of the International Journal of Molecular Sciences.
Thank you for your time and consideration,
Monika Abramiuk
Round 2
Reviewer 1 Report
Overall, the authors answered to my conerns.
I think that the results are not a 'revolution' in the field, bu that the authors gave a sound information on their results. This does not make urocortin a central marker of the disease. I'd like some more comments in the legend of figure 8 (what is the difference between the upper and the lower panel, why the endometriosis gland is labeled only in the upper panel?
Author Response
Dear Reviewer,
We would like to thank You for taking the time to read the revised version of our manuscript. We followed the suggestions of Reviewers and tried to fulfill them. All changes are marked red, in the updated version of the manuscript.
Here are the point-by-point answers.
- Overall, the authors answered to my concerns.
- We are pleased that through our answers, we were able to dispel your doubts regarding this paper.
- I think that the results are not a 'revolution' in the field, but that the authors gave a sound information on their results. This does not make urocortin a central marker of the disease. I'd like some more comments in the legend of figure 8 (what is the difference between the upper and the lower panel, why the endometriosis gland is labeled only in the upper panel?
- Thank You for this comment. We agree that urocortin understood as a single parameter is of limited importance in the diagnostic process of endometriosis, as shown by our study. Nevertheless, we believe that urocortin may have a role in the diagnosis of deep infiltrating endometriosis, which is a serious clinical problem. We sincerely hope that in the future, urocortin in combination with other markers or novel diagnostic methods may prove to be an even more useful tool.
- When it comes to Figure 8, during the editing process, we decided to replace the first histological picture we added, with another one. Therefore, Figure 8 includes only one panel (mentioned by You as the lower one). This change is visible in Microsoft Word (docx) format. Thank You for pointing out the lack of an arrow on the finally attached version of the histological picture. We have added an arrow, as well as developed the description of the histological image to make it more understandable for the readers.
Again, we would like to thank the Reviewer for his/her effort and time and we are hoping that our manuscript in its current form will fulfill the requirements of the International Journal of Molecular Sciences.
Thank You for your time and consideration,
Monika Abramiuk
Reviewer 2 Report
I dont have more comment.
Author Response
Dear Reviewer,
We would like to thank you for the informative, detailed and very satisfying reviews of our article. We believe that the excellent knowledge and commitment of the Reviewer influenced our article and made it much more relevant.
On Behalf of the authors,
Monika Abramiuk